# Chemotaxis response and age-stage, two-sex life table of the *Cheilomenes sexmaculata* (Fabricius) (Coccinellidae: Coleoptera) against different aphid species

**Hafiz Muhammad Safeer[1], Aimen Ishfaq[1], Adeel Mukhtar[1], Muazzama Batool[1], Syed Muhammad Zaka[ID][1]\*, Alia Tajdar[2], Ahmad Saood[1], Zuraiz Ali Shah[1], Muhammad Shah Zaib[1], Khalid Abbas[1], Muhammad Usama Altaf[1]**

**1** Faculty of Agricultural Sciences and Technology, Department of Entomology, Bahauddin Zakariya University, Multan, Pakistan, **2** College of plant protection, China Agricultural University, Beijing, China

\* zaka_ento@bzu.edu.pk

**Data Availability Statement:** All relevant data are within the manuscript and its Supporting Information files.

## Abstract

The *Cheilomenes sexmaculata* (Fabricius) (Coleoptera: Coccinellidae), is one of the most beneficial and identifiable predators of numerous soft-bodied and sucking insect pests of several crops. Biological parameters and olfactory response of *C. sexmaculata* were investigated under laboratory conditions by providing three different aphid species i.e., mustard aphid (*Lipaphis erysimi* Kaltenbach), citrus black aphid (*Toxoptera citricida* Kirkaldy), and peach aphid (*Diuraphis noxia* Kurdjumov) as a food source. The developmental period of immature stages of *C. sexmaculata* was shorter on *D. noxia* as compared to other aphid species. The adult longevities were longer on *D. noxia* and *T. citricida* while shorter on *L. erysimi*. Female fecundity was highest on *D. noxia* while lowest on *L. erysimi*. Life table parameters i.e., intrinsic rate of increase ($r$), finite rate of increase ($\lambda$), net reproductive rate ($R_0$), and gross reproductive rate ($GRR$) were maximum on *D. noxia* while minimum on *L. erysimi*. The mean generation time *C. sexmaculata* was 20.90, 23.69, and 26.2 days on *D. noxia*, *L. erysimi*, *T. and citricida*, respectively. These findings were further confirmed from the olfactory experiment where *D. noxia* proved to be the most preferred prey. This study provides necessary information for mass-rearing of *C. sexmaculata*.

## Introduction

There are many agricultural and horticultural crops that are susceptible to aphid infestations (Hemiptera: Aphididae) [1]. It is widely distributed in East Asia, North America [2], Europe [3], and Pakistan [4]. Both nymphs and adults [5] of the aphid have great economic importance as they cause direct damage to all plant parts by sucking cell sap [6], secret honeydew, [7] on which sooty mold develops and cause indirect damage by injecting toxic saliva along with different viruses (Pea enation mosaic virus and Bean leafroll virus) to plants [8,9]. Aphids feeding on host plants cause various losses like stunted growth, drying of leaves, and reduction in photosynthesis, which ultimately results in a yield reduction [10]. Aphids can complete more than 10 generations in a year on different hosts [11].

**Funding:** The author(s) received no specific funding for this work.

**Competing interests:** The authors have declared that no competing interests exist.

There are different control methods (cultural, mechanical, biological, and chemical) used to suppress the population of aphids in the field. Farmer community is vastly depends on chemical control for its control, in which different types of insecticides like flonicamid, sulfoxaflor, and afidopyropen are used for the management of aphids [12]. Insecticidal control is not only expensive but also harmful to the natural environment and causes several health problems [13]. Continuous applications of pesticides develop phytotoxicity in plants and destroy natural enemies such as predators, parasitoids, microorganisms, and pollinators [14,15]. There is a dire need to focus on other control methods. The use of biological control agents for the suppression of the aphid population is the best alternative control tactic [16]. The Coccinellids are efficient predators among other predators having resistance to several insecticides [17].

It is thought that coccinellids live in a variety of environments and on a wide range of hosts. A predatory coccinellid's importance goes beyond the fact that the larvae and adults eat many phytophagous soft-bodied arthropods that are destructive to agro- and forest ecosystems, including whiteflies, aphids, thrips, jassids, mealybugs, psyllids, leafhoppers, scale insects, and mites [18]. The predatory beetles, *C. sexmaculata*, are found in all Asian countries. The zigzag beetle is the most common species in Pakistan [19], India, [20], China [21], Indonesia [22], and Japan [23].

Various aphid species significantly influenced the biology and predatory efficiency of *C. sexmaculata* [24]. Furthermore, different aphid species contributed significantly to predation and life cycle attributes of *C. sexmaculata* [25]. Likewise, other coccinellid beetles have also showed significantly similar response against different prey species [26]. There was a lot of previous research published to evaluate the effect of different aphid species on life tables and the biology of coccinellids [27,28]. However, the work related to the biology and fitness of *C. sexmaculata* on different hosts by using two sex life tool was lacking. The two-sex life table provides the fitness and population parameters of both male and female insect as compared to traditional life table which describes only the female sex of an insect. Therefore, this study was performed to determine the fitness and population parameters of *C. sexmaculata* on different hosts by using two-sex life table tool. Further, olfactory studies were used extensively to evaluate the chemotaxis responses of the ladybird beetle towards its prey [29–31] previously no record of this aspect of *C. sexmaculata* has been observed. Based on the importance of aphids and ladybird beetles, the current study aimed to evaluate the developmental and behavioral responses of *C. sexmaculata* using different species of aphids.

## Material and methods

### Collection and mass-rearing of predator

The field population of *C. sexmaculata* was collected by using an aspirator and camel hairbrush from *Calotropis procera* present around the Agricultural fields, Bahauddin Zakariya University, Multan, Pakistan. Field collected adults and larvae were placed separately in rearing plastic jars (8×8×14 cm) reared till homogenous population was achieved under control conditions (25±1˚C and 70±2% R.H. and 16L:8D h) [32]. Different nymphal instars of Aphids were used as a food source for both larvae and adults. The jars were closed with the muslin cloth (440 μm) for ventilation and to avoid contamination. Eggs were collected from adults rearing jars and placed in a petri dish (6cm diameter), having moist filter paper at the base [33].

### Collection and rearing of prey

The collection of three different aphid species Mustard aphid (*Lipaphis erysimi* Kaltenbach), Citrus black aphid (*Toxoptera citricida* Kirkaldy), and Peach aphid (*Diuraphis noxia* Kurdjumov) was done from their respective hosts i.e., *Brassica campestris*, *Citrus clementina*, and

*Triticum aestivum*. These aphid species were reared on their respective host plants separately in cages (43.75×43.75×50 cm) under laboratory-controlled conditions (26±2°C, 65±5% R.H and L14:D10 photoperiod) and fresh diets were provided on daily basis [18].

## Biological parameters

Eggs of *C. sexmaculata* were taken from a homogeneous population and placed singly in the petri dish (6cm diameter). After hatching, three aphid species were provided throughout their whole life and each aphid species was considered as a treatment. There were thirty replications in each treatment. The sufficient but similar number of aphids were released in each replication on daily basis till the formation of pupa. Different larval instars of *C. sexmaculata* were identified by the presence of exuvium. After the emergence of adults, five pairs (each pair represent one replication) of each treatment were placed in plastic jars (8×8×14 cm) separately. Pre-oviposition, oviposition, post-oviposition, adult survival rate, male and female longevity were recorded [34].

## Olfactometer experiment

The olfactory response of all four instars and adults (male and female) of *C. sexmaculata* against two different aphid species (*D. noxia* and *L. erysimi*) was determined by using a four-arm olfactometer as described by Vet et al [35]. The olfactory chamber was connected with four arms (each arm consists of three flasks having odor source, water and charcoal, respectively, followed by an airflow meter) and a hole in the center for suction, the same hole was also used for entry of insect. The suction pump was connected at the central hole of the olfactory chamber and the flow of air was adjusted by air flow meter at 4 kpa/min. The olfactory chamber was divided into four regions (A, B, C and D), each region representing the specific odor received from a specific arm. The light source (12 KW LED bulb) was also hung at the center of the olfactory chamber [36]. A black cloth was used around the whole olfactory apparatus to ensure the uniform light condition.

The larval instars (1st, 2nd, 3rd & 4th) and adults (male and female) of *C. sexmaculata* were placed singly in the olfactory chamber through central opening and monitored for 10 minutes [37]. The retention time was recorded by using a stopwatch and the number of entries by visual counting. Data recording was started after 15 seconds (settlement time). The individuals who spent more than 50% time in the central region of the olfactory chamber were discarded. Three replications were performed of each instar as well as adults. The insect was considered as a choice insect when it remained 15 seconds or more in the order source region. Moreover, the direction of the apparatus was changed to 90° after every replication to remove the possible side effects on the insect behavior. The apparatus was washed with detergents and 70% ethanol after each replication and kept for drying to remove the contamination [29].

## Statistical analysis

The biological parameters and olfactory response of larvae and adults of *C. sexmaculata* were analyzed by one-way analysis of variance (ANOVA) with the help of the statistical package SAS (Version 8.0) [38]. The means of different treatments were compared by using the Least Significant Difference (LSD) at 5% probability.

TWO SEX-MS Chart was used to compute different population parameters of *C. sexmaculata* [39]. The bootstrap technique of TWO SEX-MS Chart with 100,000 replications were performed to minimize the variations in the results [40]. The recorded data were used to analyze the age-stage specific fecundity ($f_{xj}$, where x = age in days and j = stage), age-stage–specific survival rate ($S_{xj}$), age-stage reproductive value ($V_{xj}$), age-stage life expectancy($e_{xj}$), age-specific

fecundity ($m_x$), age-specific survival rate ($l_x$), age-specific net maternity ($l_x m_x$), and life table parameters like, $\lambda$, finite rate of increase $r$, intrinsic rate of increase $R_0$, net reproductive rate; and $T$, the mean generation [41].

In the age-stage, two-sex life table, the age-specific survival rate ($l_x$), age-specific fecundity ($m_x$), net reproductive rate ($R_o$), finite rate of increase ($r$), finite rate ($\lambda$), mean generation time ($T$) and life expectancy ($e_{xj}$) were calculated by following equations;

$$l_x = \sum_{j=1}^{k} S_{xj} \tag{1}$$

$$m_x = \frac{\sum_{j=1}^{k} S_{xj} f_{xj}}{\sum_{j=1}^{k} S_{xj}} \tag{2}$$

Where $k$ is the number of stages.

$$R_0 = \sum_{x=0}^{\infty} l_x m_x \tag{3}$$

where the net reproductive rate is the mean number of offspring laid by the individual during its entire life span.

$$\sum_{x=0}^{\infty} e^{-r(x+1)} l_x m_x = 1 \tag{4}$$

Where Iterative bisection method was used to estimate and corrected the intrinsic rate of increase (r) with the Euler–Lotka equation [42].

$$\lambda = e^r \tag{5}$$

$$T = In^{R_0}/_r \tag{6}$$

Where mean generation time is defined as the length of time that a population needs to increase to $R_0$-fold of its population size at the stable age-stage distribution.

$$e_{xj} = \sum_{i=x}^{\infty} \sum_{y=j}^{\beta} s\prime_{iy} \tag{7}$$

Where life expectancy ($e_{xj}$) is the length of time that an individual of age $x$ and stage $j$ is expected to live [43–45].

## Results

### Biological parameters

The developmental period of each immature stage and adult longevity of *C. sexmaculata* fed on three different aphid species (Table 1). The developmental period of the eggs was statistically the same for all tested aphid species. The developmental period of the first larval instar was significantly shorter (F = 19.35; DF = 2,87; P<0.0001) when reared on *D. noxia* (0.93 days), followed by 1.60 days when fed on *T. citricida*, while the longest (2.50 days) duration was observed when fed on *L. erysimi*. The second instar larvae completed its development significantly (F = 14.21; DF = 2,87; P<0.0001) shorter (00 days) on *D. noxia*, while reared on *T. citricida* completed its development in 1.73 days and the longest development time (2.67 days) was observed on *L. erysimi*. A shorter developmental period (1.40 days) was observed in third instar larvae when *D. noxia* was provided as food, followed by 1.90 days on *T. citricida* and longer development time i.e., 2.47days on *L. erysimi* (F = 3.24; DF = 2,87; P = 0.0463). Fourth instars and pupae of the *C. sexmaculata* shared the same statistical rank on all aphid species.

**Table 1. Development period (mean ± SE) of *M. sexamaculatus* fed on three aphid species.**

| Developmental Stages/ Ovipositioning | *Lipaphis erysimi* | *Toxoptera citricida* | *Diuraphis noxia* | Statistical parameters | | |
|---|---|---|---|---|---|---|
| | | | | F-value | df**, Edf*** | P-value |
| Eggs(days) | 1.80a ± 0.11 | 1.73a ± 0.13 | 1.73a ± 0.13 | 0.1 | 2,87 | 0.9077 |
| First instar(days) | 2.50a ± 0.24 | 1.60b ± 0.16 | 0.93c ± 0.08 | 19.35 | 2,87 | <0.0001 |
| Second instar(days) | 2.67a ± 0.31 | 1.73b ± 0.19 | 1.00c ± 0.11 | 14.21 | 2,87 | <0.0001 |
| Third instar(days) | 2.47a ±0.37 | 1.90ab ± 0.22 | 1.40b ± 0.20 | 3.24 | 2,87 | 0.0463 |
| Fourth instar(days) | 2.00± 0.33 | 1.70 ± 0.26 | 1.60 ± 0.21 | 0.64 | 2,87 | 0.5317 |
| Pupa(days) | 1.63 ± 0.29 | 1.93 ± 0.30 | 1.53 ± 0.22 | 0.62 | 2,87 | 0.5409 |
| Total period from egg to pupae(days) | 13.07a ± 1.22 | 10.53ab ± 1.00 | 8.20b ± 0.68 | 6.00 | 2,87 | 0.0043 |
| Male longevity(days) | 11.11b ±1.20 | 26.22a ± 0.94 | 26.00a ± 0.85 | 112 | 2,24 | <0.0001 |
| Female longevity(days) | 12.13b ± 1.15 | 32.0a ± 0.66 | 32.25a ± 0.70 | 158.18 | 2,21 | <0.0001 |
| Pre-oviposition period(days) | 4.00b ± 0.42 | 6.25a ± 0.59 | 5.13ab ± 0.71 | 9.00 | 2,35 | 0.0031 |
| Oviposition period(days) | 5.00b ± 0.53 | 9.12a ± 0.29 | 10.25a ± 0.72 | 29.42 | 2,35 | <0.0001 |
| Post-oviposition period(days) | 3.13b ± 0.72 | 16.50a ± 0.88 | 16.63a ± 0.54 | 93.31 | 2,35 | <0.0001 |
| Fecundity (eggs/female) | 64.88b ± 9.97 | 216.6a ± 15.05 | 265.91a ± 33.41 | 18.55 | 2,35 | <0.0001 |

*Mean followed by different letters in the same row are significantly different ($P < 0.05$).

**df stands for Degree of freedom.

***Edf stands for Error degree of freedom.

All three aphid species have significant (F = 6.00; DF = 2,87; P = 0.0043) influenced on the total duration of immature stages. The total developmental duration of immature stages was longer on *L. erysimi* (13.07 days) but shorter on *D. noxia* (8.20 days).

A significant difference in adult longevity of males and females was observed when different aphid species were provided to them as food. Males and females lived significantly shorter (F = 112; DF = 2,24; P = 0.0001 and F = 158.18; DF = 2,1; P = 0.0001, respectively) when fed on *L. erysimi* (11.11 and 12.13 days, respectively) and longer on *D. noxia* (26days) and *T. citricida* (32days), respectively. When the beetles fed on different aphid's species showed the significant effect on pre-oviposition period (F = 9.00; DF = 2,35; P = 0.0031) and post oviposition period (F = 93.31; DF = 2,87; P = 0.0001). The oviposition period was maximum (10.25 and 9.12 days) when *C. sexmaculata* fed on *D. noxia* and *T. citricida*, respectively, while the minimum was recorded on *L. erysimi* (5.00 days) (F = 29.42; DF = 2,35; P<0.0001).

Fecundity of *C. sexmaculata* varied significantly (F = 18.55; DF = 2,35; P<0.0001) on different aphid species. Maximum fecundity was recorded when *D. noxia* (265.91eggs/female) and *T. citricida* (216.60eggs/female) were given as food while minimum fecundity was recorded on *L. erysimi* (64.88 eggs/female) (Table 1).

Life table parameters i.e., r, λ, and *R*o (0.21 d$^{-1}$,1.24 d$^{-1}$ and 97.5 offspring, respectively) were higher on *D. noxia* as compared to other tested aphid species. Mean generation time (*T*) was higher (26.20 $^{\text{d-1}}$) on *T. citricida* than on *L. erysimi* (23.69 d$^{-1}$) while the lowest was observed on *D. noxia* (20.90 d$^{-1}$). The maximum gross reproductive rate (*GRR*) of the *C. sexmaculata* was 146.2 offspring when fed on *D. noxia* followed by 114 and 63.08 offspring on *T. citricida* and *L. erysimi* respectively (Table 2).

The eggs and second instar exhibited the survival probability (S$_{xj}$) 1 and 0.8667, respectively against all provided aphid species. Similar survival probability of first and third instars of *C. sexmaculata* was observed i.e., 0.8667 and 0.8333, respectively, on *D. noxia* and *T. citricida* while on *L. erysimi* survival probability was 0.93 and 0.8, respectively for both first and third instars. In the case of the fourth instar survival probability (0.8333) was observed on *D. noxia* followed by 0.8 on *T. citricida* while 0.7 was observed on *L. erysimi*. The pupae of

**Table 2. Life table parameters of *M. sexamaculatus* reared on three tested aphid species.**

| Parameters | Values of life table parameters on aphid species | | |
|---|---|---|---|
| | *Lipaphis erysimi* | *Toxoptera citricida* | *Diuraphis noxia* |
| $r$ (d$^{-1}$) | 0.12 c ± 0.01 | 0.16 b ± 0.03 | 0.21 a ± 0.02 |
| $\lambda$ (d$^{-1}$) | 1.13 b ± 0.03 | 1.17 b ± 0.02 | 1.24 a ± 0.01 |
| $Ro$ (Offspring individual $^1$) | 17.30 c ± 2.34 | 72.2 b ± 2.12 | 97.50 a ± 3.22 |
| $T$ (d) | 23.69b ± 0.32 | 26.20 a ± 0.33 | 20.90 c ± 0.32 |
| $GRR$(Offspring) | 63.08 c ± 11.23 | 114 b ± 13.22 | 146.2 a ± 19.23 |

*$r$ = intrinsic rate of increase, $\lambda$ = finite rate of increase, $Ro$ = net reproductive rate $T$ = mean generation time, $GRR$ = the gross reproductive rate.

*C. sexmaculata* exhibited the survival probability (0.7333) on *D. noxia* followed by 0.6333 and 0.6 on *T. citricida* and *L. erysimi*, respectively. A similar trend of survival probability was observed in females i.e., 0.3667, 0.3333, and 0.2667 in connection with *D. noxia*, *T. citricida*, and *L. erysimi*, respectively. Survival probabilities (0.3) of males were the same with all tested aphid species (Fig 1).

*Cheilomenes sexmaculata* showed the highest survival rate (*lx*) on *D. noxia*and lowest on *L. erysimi*. The age-stage-specific female fecundity (*fx7*) of *C. sexmaculata* was maximum on *T. citricida* (34.1 eggs at the age 27 days), but minimum on *L. erysimi* (16 eggs at the age 28 days). Similarly, age-specific fecundity (*mx*) and Age-specific net maternity (*lxmx*) highest on *T. citricida*, but lowest on *L. erysimi* (Fig 2).

The value of age-stage–specific reproductive rates (*Vxj*) was highest in thecase of *T. citricida* (116.44 eggs at the age 23 days) and lowest on *L. erysimi* (48.02 eggs at the age 28 days) (Fig 3).

Maximum life expectancy (*exj*) 33.78 was observed at the pupal stage after 10 days in the case of *T. citricida*. In the case of *D. noxia*, a maximum life expectancy 34.82 was observed in the female stage at age 7 days. Maximum life expectancy 20.76 was observed in the third larval instar at age 5 days in the case of *L. erysimi* (Fig 4).

## Olfaction

**Olfactory response of larval instars of *C. sexmaculata* against *D. noxia* and *L. erysimi*.** The first instar of *C. sexmaculata* spent significantly ($p<0.0001$) maximum time (205.0 sec) in the region having the odor of *D. noxia* as compared to blank. While, in the case of *L. erysimi*, a non-significant ($p = 0.0620$) response was observed to all four regions of the olfactometer. Significantly ($p< 0.0064$) the highest time (202.3 sec) was spent by the second instar in the region having volatiles of *D. noxia* than control. Similarly, a significant ($p = 0.0075$) response was observed against *L. erysimi*. The third instar of *C. sexmaculata* spent significantly ($p = 0.0215$) more time (188.0 sec) in the region where volatiles of *D. noxia* was provided than control, while in the case of *L. erysimi*, non-significant ($p = 0.1405$) attraction was observed. The fourth instar of *C. sexmaculata* spent significantly ($p = 0.0039$) maximum time (206.3 sec) in the region having the odor of *D. noxia* as compared to blank. Moreover, a non-significant ($p = 0.0878$) response was observed in the case of *L.erysimi* (Fig 5).

**Olfactory response of male and female of *C. sexmaculata* against *D. noxia* and *L. erysimi*.** The male of *C. sexmaculata* spent significantly ($p = 0.004$) maximum time (213.3 sec) in the region having the odor of *D. noxia* as compared to blank while, in the case of *L. erysimi*, non-significant ($p = 0.1094$) response was observed. The significantly ($p<0.0001$) maximum time (239.0 sec) spent by the female of *C. sexmaculata* in the region having the odor of *D. noxia* as compared to blank while, in the case of *L. erysimi*, non-significant ($p = 0.0753$) response was observed (Fig 6).

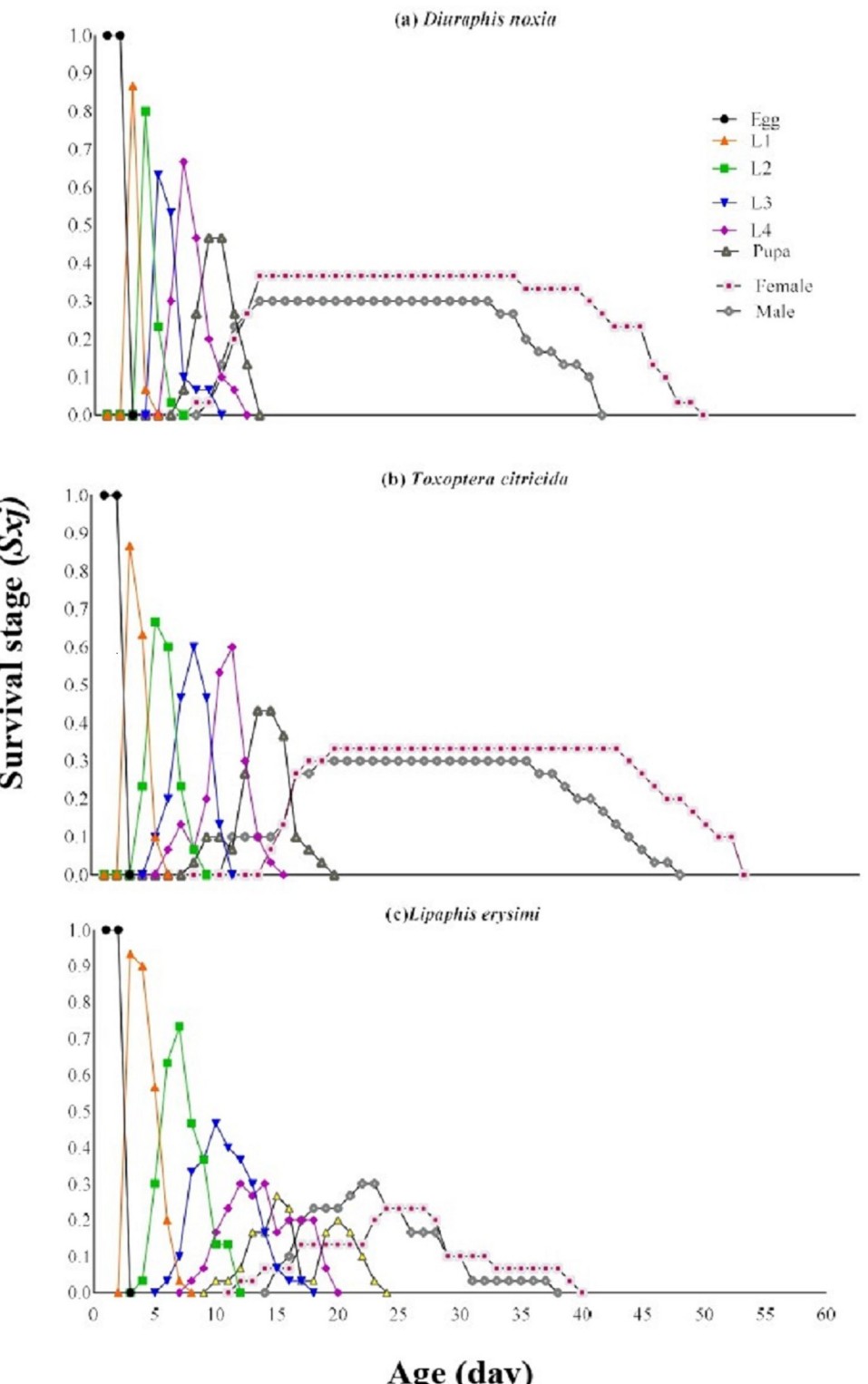

**Fig 1. Age-stage–specific survival rate ($S_{xj}$) of *C. sexmaculata* fed on three aphid species.**

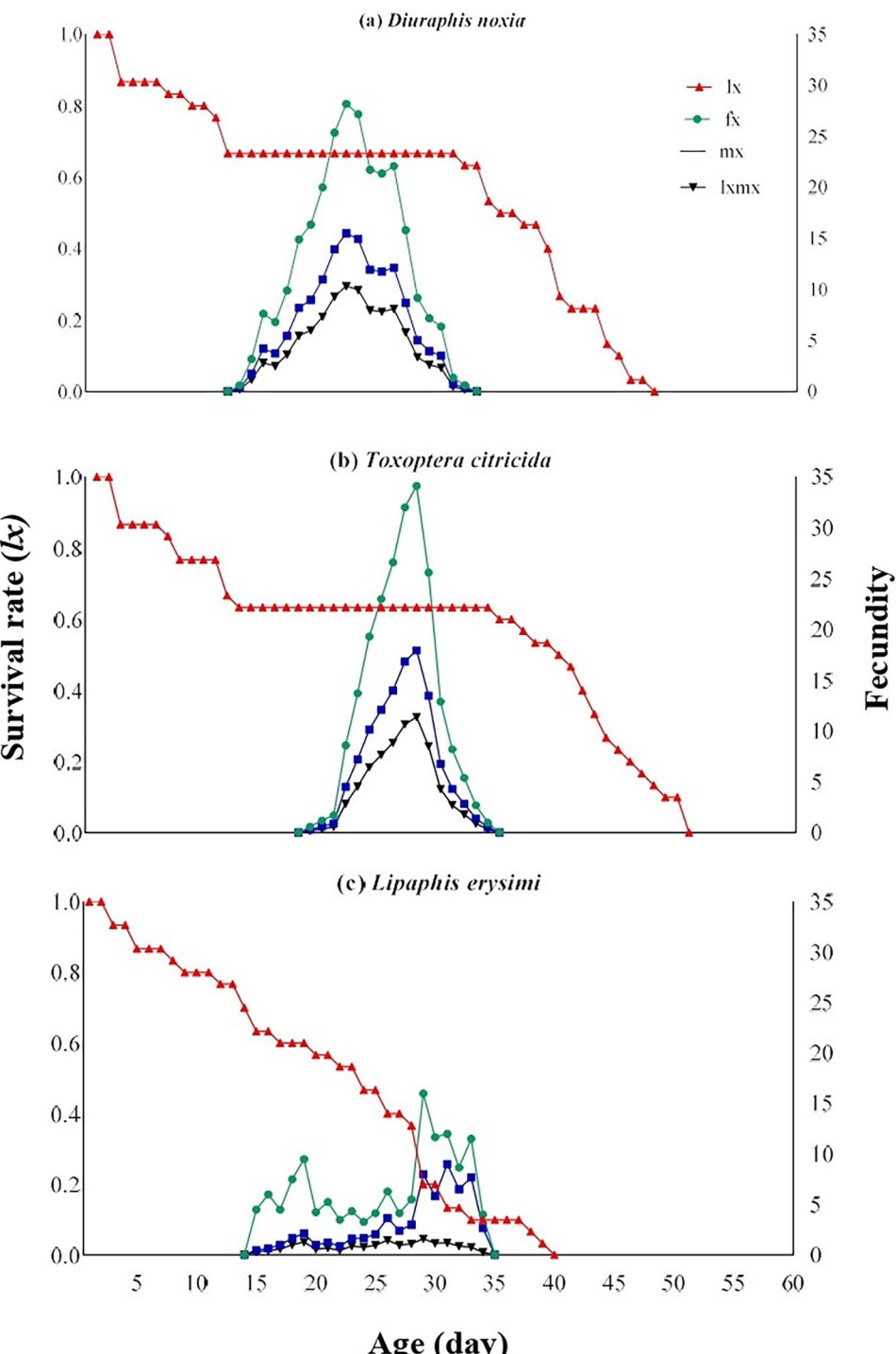

**Fig 2. Age-specific survival rate (*lx*), age-stage–specific fecundity (*fxj*), age-specific fecundity (*mx*), and age-specific maternity (*lxmx*) of *C. sexmaculata* fed on three aphid species.**

## Discussion

The purpose of this study was to examine how different aphid species affect the development, survival rate, and fecundity of *C. sexmaculata*. The outcomes showed that the availability and

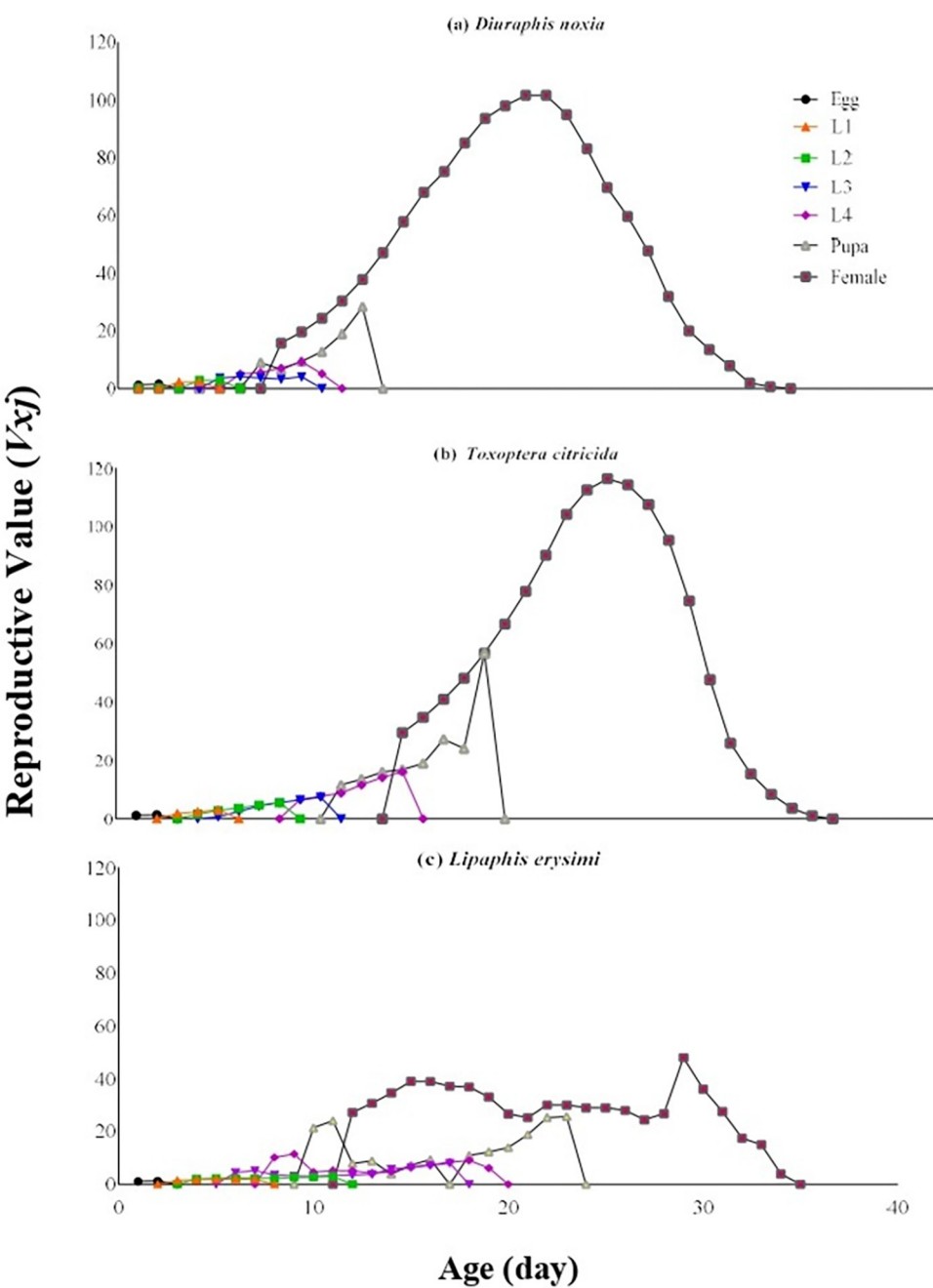

**Fig 3. Age-stage–specific reproductive rate ($V_{xj}$) of *C. sexmaculata* fed on three aphid species.**

quality of prey influenced the development of *C. sexmaculata*. These findings are in consistent with the biological study of *C. sexmaculata* on different aphid species in which the nature and quality of the prey change the feeding behavior of predators and alter their development, survival rate, and fecundity [19]. The findings of current research evidenced that the development period was delayed in the case of *L. erysimi* as compared to other aphid species, which showed that prey quality effects predator development. A similar finding was observed in a study where Predator development is delayed by the deficient quality and inadequate amount of

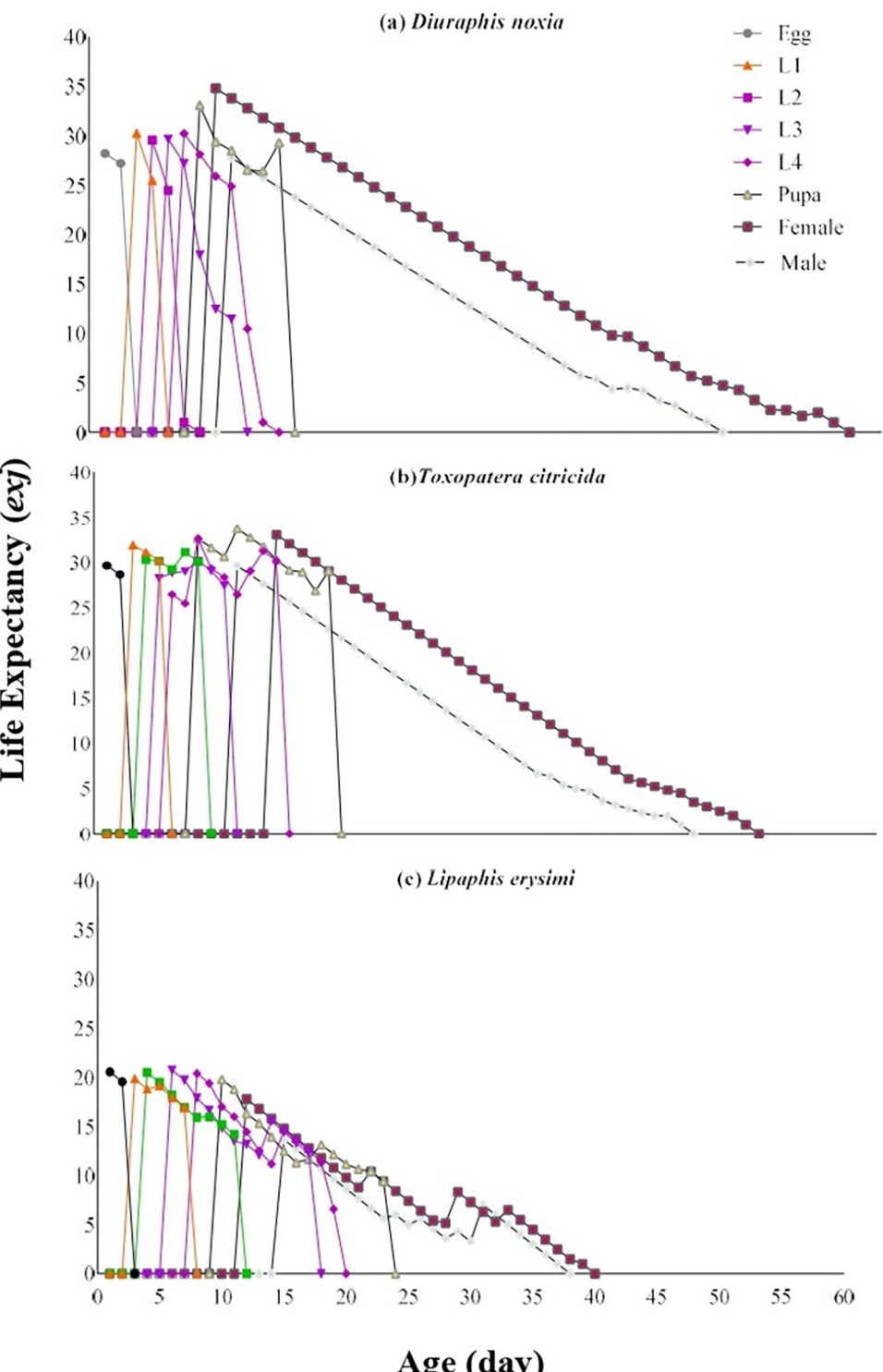

**Fig 4. Age-stage–specific life expectancy (*exj*) of *C. sexmaculata* fed on three aphid species.**

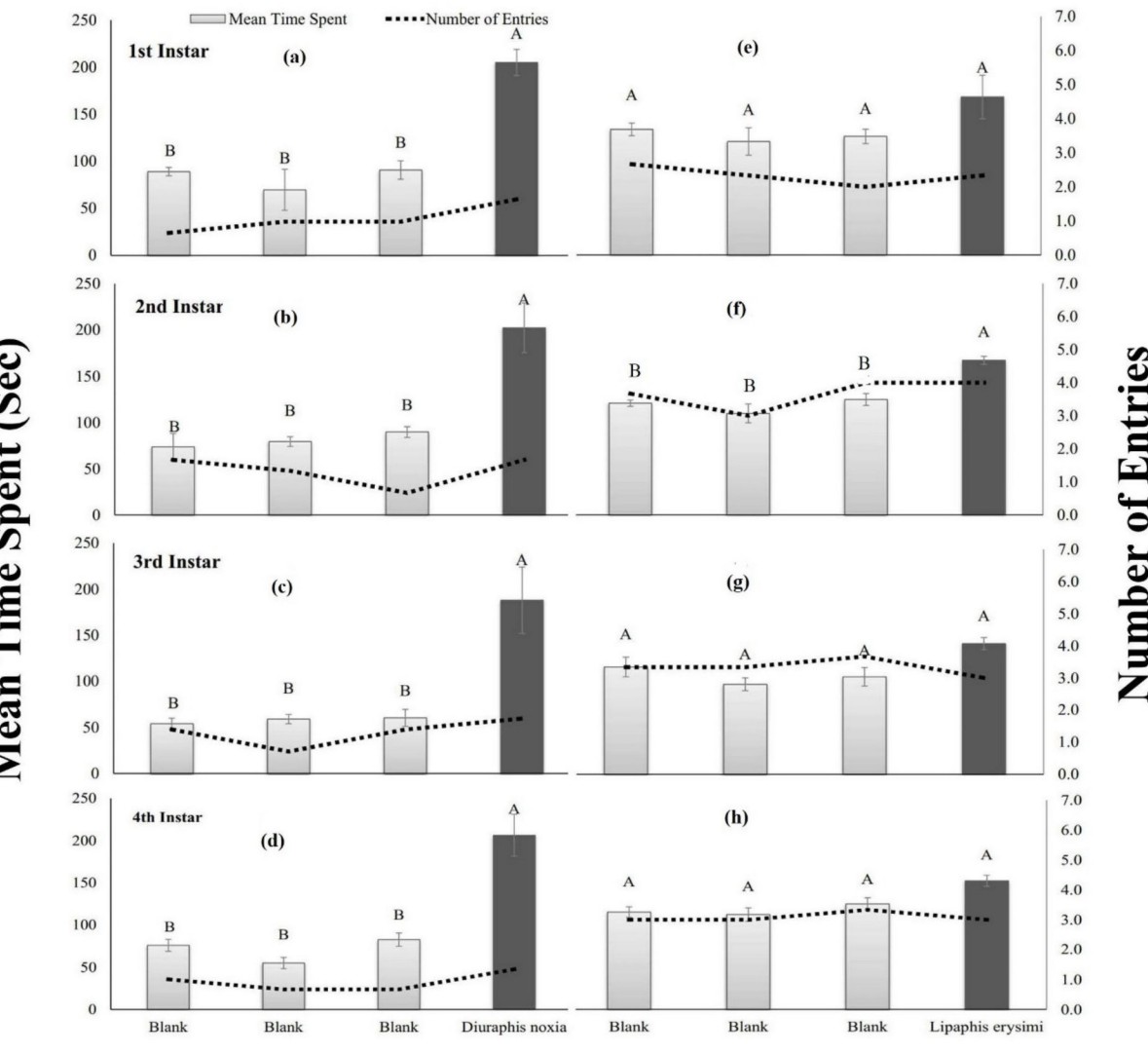

**Fig 5.** Mean time spent (sec) and the number of entries in each region of a four-arm olfactometer by different instars of *C. sexmaculata* exposed to *D. noxia* (a, b, c & d) and *L. erysimi* (e, f, g & h).

prey, while predator development is encouraged by the nutritious quality and adequate amount of prey [46]. The developmental period of immatures of coccinellid fluctuated greatly by feeding on different aphid species, this is comparable with the findings showed that the longest preadult developmental period of *C. septempunctata* was observed on *A. craccivora* while the shortest was seen on *M. persicae* [27]. The results of the present study showed that adults' longevities were longer on *D. noxia* while shorter on *L. erysimi*. It correlates to a biological study of *C. sexmaculata* on different seven aphid species. The longer adult longevity of predatory beetle was observed on *A. craccivora* while shorter on *L. erysimi* than other tested aphid species [26]. A shorter developmental period of *C. sexmaculata* was observed on *L. erysimi* in the present study. These findings are conflicted with the study that was performed on *C. septempunctata* and proved that the longer developmental period was observed on *L. erysimi* while shorter on *M. rosae* [47]. The variations occurred in the results of different studies due to the different biotic (predator, prey and plant species) and abiotic (mass-rearing conditions of predator and prey) factors.

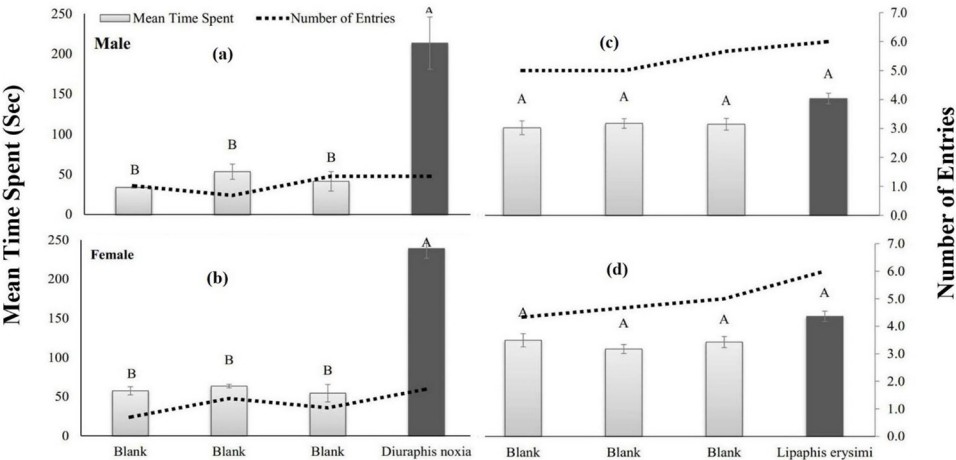

**Fig 6.** Mean time spent (sec) and the number of entries in each region of a four-arm olfactometer by different instars of *C. sexmaculata* exposed to *D. noxia* (a & b) and *L. erysimi* (c & d).

The quantity and quality of aphid species have a direct effect on female fecundity. The highest fecundity was observed when *D. noxia* was provided as food and the lowest fecundity was observed on *L. erysimi*. Previous studies also proved that the dietary quality of prey influenced predator fecundity [48]. In the current study, the highest female fecundity was observed on *D. noxia* while the lowest was on *L. erysimi*. The findings of the current study are correlated with the previous study indicated that the maximum lifetime fecundity of *Anegleis cardoni* was found on *A. gossypii* and the minimum on *L. erysimi* [49].

There were many difficulties associated with the traditional life tables that explained the female population but neglected the male populations and stage variation among individuals and both sexes. A modern life table (age-stage two-sex life table) was applied in the present study to overcome the difficulties (the variations among age-specific fecundity and age-specific survival rate) connected to traditional life tables. Life table parameters ($l_x$ and $m_x$) were used for the consideration of the survival rate of male and stage variation among individuals. A detailed description of the issues and errors with female age-specific life tables was provided [43,50].

Life table parameters are greatly affected by various factors i.e., the quality and nature of host plant and prey species, and controlled conditions of the laboratory [51,52]. The outcomes of the present study exhibited that maximum $R_0$, $r$, $\lambda$, and *GRR* parameters were recorded on *D. noxia* and minimum on *L. erysimi*, while *T* was maximum on *T. citricida* followed by *L. erysimi* and minimum on *D. noxia*. These observations are similar to the study that outcomes revealed that $R_0$, $r$, $\lambda$, and *GRR* were highest on *A. gossypii* and lowest on *L. erysimi* while *T* was maximum on *L. erysimi* and minimum on *A. gossypii* [19,49].

The survival rate of *C. sexmaculata* was highest when fed on *D. noxia*, according to the age-stage-specific survival rate ($S_{xj}$) curves. The findings are comparable to the study of *C. sexmaculata* in which the highest survival rate was also recorded on *D. noxia* than other aphid species [19]. The highest age-stage-specific female fecundity and Age-specific survival rate of *C. sexmaculata* were observed on *D. noxia* while the lowest age-stage-specific female fecundity and Age-specific survival rate were recorded on *L. erysimi*. The results of the previous study correlated with current research that *L. erysimi* exhibited the lowest age-stage-specific female fecundity with *H. convergens*, while the age-specific survival rate of *H. convergens* was maximum on *L. erysimi*, which is contrary to the present result [28]. The difference in the results depends on the species of predator and prey. The age-stage–specific reproductive rate ($v_{xj}$) provides precise

information of an individual's contribution to population increase in the future at age *x* and stage *j*. The age-stage–specific reproductive rate ($v_{xj}$) was highest in the case of *T. citricida* and lowest on *L. erysimi*. Similar results were found in the study of *H. convergens* on different aphid species [28]. The results showed that the life expectancy of individuals decreased with the increase of age. This is consistent with a study on *C. flavocapitis* under laboratory conditions, which found that life expectancy declined with age [53] The age-stage two-sex life table described the biological parameters of all stages of *C. sexmaculata*.

## Conclusion

This study concluded that prey species significantly affected the biological parameters of *C. sexmaculata*. and *D. noxia* is proved to be most suitable host for the mass rearing of *C. sexmaculata* under laboratory conditions. *Coccinella sexmaculata* can enhance the overall control of aphids when incorporated into an IPM strategy. IPM combines multiple pest control methods, including biological, cultural, and chemical approaches, to minimize the reliance on pesticides and promote sustainable pest management practices.

## Supporting information

**S1 Raw data.**
(ZIP)

## Acknowledgments

The authors wish to acknowledge the Department of Entomology, for coordinating in the present research work and undergraduate students for helping in experiments. Further, authors are thankful to undergraduate students for helping during research work.

## Author Contributions

**Conceptualization:** Syed Muhammad Zaka.

**Data curation:** Hafiz Muhammad Safeer, Aimen Ishfaq, Adeel Mukhtar, Ahmad Saood, Zuraiz Ali Shah, Muhammad Shah Zaib, Khalid Abbas.

**Formal analysis:** Hafiz Muhammad Safeer, Aimen Ishfaq, Muazzama Batool, Alia Tajdar.

**Investigation:** Hafiz Muhammad Safeer, Adeel Mukhtar, Syed Muhammad Zaka, Muhammad Usama Altaf.

**Methodology:** Hafiz Muhammad Safeer, Aimen Ishfaq, Syed Muhammad Zaka, Alia Tajdar, Ahmad Saood, Zuraiz Ali Shah, Muhammad Usama Altaf.

**Project administration:** Syed Muhammad Zaka.

**Resources:** Alia Tajdar.

**Software:** Muhammad Shah Zaib, Khalid Abbas.

**Supervision:** Syed Muhammad Zaka.

**Visualization:** Aimen Ishfaq.

**Writing – original draft:** Hafiz Muhammad Safeer.

**Writing – review & editing:** Aimen Ishfaq, Muazzama Batool, Syed Muhammad Zaka, Alia Tajdar.

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
