## [Decision Letter · Decision Letter 0]

10 Jan 2023

PONE-D-22-16646Chemotaxis response and age-stage, two-sex life table of the Cheilomenes sexmaculata (Fabricius)  (coccinellidae: coleoptera) against different aphid speciesPLOS ONE

Dear Dr. Zaka,

Thank you for submitting your manuscript to PLOS ONE. After careful consideration, we feel that it has merit but does not fully meet PLOS ONE’s publication criteria as it currently stands. Therefore, we invite you to submit a revised version of the manuscript that addresses the points raised by the two reviewers during the review process.

We look forward to receiving your revised manuscript.

Kind regards,

Nicolas Desneux

Academic Editor

PLOS ONE

Journal Requirements:

Reviewers' comments:

Reviewer's Responses to Questions

**Comments to the Author**

1. Is the manuscript technically sound, and do the data support the conclusions?

Reviewer #1: Yes

Reviewer #2: Partly

2. Has the statistical analysis been performed appropriately and rigorously? 

Reviewer #1: Yes

Reviewer #2: Yes

3. Have the authors made all data underlying the findings in their manuscript fully available?

Reviewer #1: Yes

Reviewer #2: No

4. Is the manuscript presented in an intelligible fashion and written in standard English?

Reviewer #1: Yes

Reviewer #2: Yes

5. Review Comments to the Author

Reviewer #1: I appreciate the Editor to give me a chance to review an interesting and valuable paper. I would like add few points in the manuscript entitled "Chemotaxis response and age-stage, two-sex life table of the Cheilomenes sexmaculata (Fabricius) (Coccinellidae: Coleoptera) against different aphid species". In my opinion, this paper has a good potential to be published in the journal. I have indicated my comments directly in the attached annotated manuscript.

Reviewer #2: In this study, the authors investigated the biological parameters and olfactory response of Cheilomenes sexmaculata (Fabricius) (Coleoptera: Coccinellidae) under laboratory conditions by providing three different aphid species i.e., mustard aphid (Lipaphis erysimi), citrus black aphid (Toxoptera citricida), and peach aphid (Diuraphis noxia) as a food source. Although the topic is worth of interest and the methods used are standard, there is insufficient data to support its main conclusion. I encourage the authors to work more and notably to assess the population projection using TIMING-MSChart computer program. Besides, the authors should check the predation rate using age-stage, two-sex life table. The manuscript needs careful proofreading and revision. Grammar mistakes are undermining the significance of this study. Therefore, I think it cannot be accepted in its current form in PONE. I recommend a major revision, in which the following key points should be addressed.

Major Points:

- The authors should add concluding lines. It should be more specific and striking.

- The introduction section is not coherent. There are many useless sentences and lack key information. It should be rewritten completely. The author needs to add info about Age-stage, two-sex life table approach by TWO-SEX MS Chart. What is the difference between this technique and the traditional technique?

-In results, please follow the correct pattern for writing the statistical values. For example in L168, you may replace (F=19.35; F2,87; P<0.0001) by (F=19.35, DF=2,87, P<0.0001). Please write the exact P value. Follow this pattern throughout the manuscript.

- It is recommended to discuss and explain what should be the appropriate policies based on the findings of the current study. Moreover, the results should be further elaborated to show how they could be used for real applications.

-Figure 1-4: I strongly suggest to replace these figures by color figures. The colors must be more contrasting, so that readers can easily understand the variations among different parameters etc.

-In statistical analysis section, the authors missed key references (Chi et al. 2022a; 2022b) that strongly support the age-stage, two-sex, life table approach.

- Chi et al. 2022a. TWOSEX-MSChart: the key tool for life table research and education. Entomologia Generalis. 42 (6): 845-849.

- Chi et al. 2022b. Innovative application of set theory, Cartesian product, and multinomial theorem in demographic research. Entomologia Generalis. 42 (6) 863-874.

- I strongly suggest the authors to check the population projection via TIMING-MSChart computer program. Construct figures of population projections, add formula of population projection statistical analysis section, add results with separate heading (such as Population Projection), and finally, discuss these results in discussion section by comparing it with recently published articles from reputable journals.

- I strongly suggest authors to check the predation rate using age-stage, two-sex life table approach, as this software precisely describes the predation rate.

Ding, H. Y., Lin, Y. Y., Tuan, S. J., Tang, L. C., Chi, H., Atlıhan, R., ... & Güncan, A. (2021). Integrating demography, predation rate, and computer simulation for evaluation of Orius strigicollis as biological control agent against Frankliniella intonsa. Entomologia Generalis, 41(2), 179-196.

- Islam, Y., Güncan, A., Fan, Y., Zhou, X., Naeem, A., & Shah, F. M. (2022). Age-stage, two-sex life table and predation parameters of Harmonia axyridis Pallas (Coleoptera: Coccinellidae), reared on Acyrthosiphon pisum (Harris)(Hemiptera: Aphididae), at four different temperatures. Crop Protection, 106029.

- Yılmaz, M., & Polat Akköprü, E. (2021). Predation rate linked to life table of Chrysoperla carnea (Stephen)(Neuroptera: Chrysopidae) fed on small walnut aphid (Chromaphis juglandicola)(Kalt.)(Hemiptera: Aphididae): with population and predation projections. Phytoparasitica, 49(2), 217-228.

- In Table 2, the authors didn’t added the standard errors, also missed the P values and different letters to show the significant differences. These are very important, I am surprised why authors missed these key information.

- The authors should add a separate conclusion section after discussion. The conclusion section should be concise and to the point.

-Correct Ref# 39:

“Chi H. TIMING-MSChart: a computer program for the population projection based on age-stage, two-sex life table. 2016” should be replace by “Chi, H. TWOSEX-MS Chart: A Computer Program for the Age-Stage, Two-Sex Life Table Analysis. 2022. Available online: http://140.120.197.173/ecology/Download/Twosex-MSChart-exe-B100000.rar (access date).” Please mention the access date (day, month, year).

- Regarding data availability, upload all raw datasets of the life table and functional response as supplementary files. Raw data files should be available as supplementary files.

- Please improve the figures quality especially the font size and colors. These figures should be more striking.

6. PLOS authors have the option to publish the peer review history of their article (what does this mean?). If published, this will include your full peer review and any attached files.

Reviewer #1: No

Reviewer #2: No

---

## [Author Response · Author response to Decision Letter 0]

28 Jun 2023

RESPONSE TO REVIEWER 1

Comment: Line number 3 “(coccinellidae: coleoptera)” 

Response: Done Accordingly. Line number 3 “(Coccinellidae: Coleoptera )”

Comment: Line number 25 Provide Authority name “(Lipaphis erysimi), citrus black aphid (Toxoptera citricida), and peach aphid (Diuraphis noxia).

Response: Done Accordingly. Line number 25-26 “(Lipaphis erysimi Kaltenbach), citrus black aphid (Toxoptera citricida Kirkaldy), and peach aphid (Diuraphis noxia Kurdjumov)”.

Comment: Line 44-45 Modify the sentence.”Persistent feeding of aphid population

 to host plant cause the different losses like stunted plant growth drying of leaves, decrease in

46 photosynthesis that results in the reduction of yield”.

Response: Done Accordingly. Line number 47-49 “Aphids feeding on host plants cause various losses like stunted growth, drying of leaves, and reduction in photosynthesis, which ultimately results in a yield reduction”.

Comment: Line number. Use single word “mealy bugs”

Response: Done Accordingly. Line number 63 “mealybugs”.

Comment: Line number 52-53. Keep either common name or Scientific name “The zigzag beetle C.

sexmaculata is the most common species in Pakistan”.

Response: Done Accordingly. Line number 71 “The zigzag beetle is the most common species in Pakistan”

Comment: Line number 55-56. Please indicate about the aphids “used to suppress the pest population in the field”.

Response: Done Accordingly. Line number 52 “used to suppress the population of aphids in the field”.

Comment: Line number 67-68. Modify the sentence “Likewise, on other coccinellid beetles, similar significant responses of predatory coccinellids against different prey species have been reported”.

Response: Done Accordingly. Line number 76-77 “Likewise, other coccinellid beetles have also showed significantly similar response against different prey species”.

Comment: Line number 82.which stage has been used as a food source “Aphids were used as a food source for both larvae and adults”.

Response: Done Accordingly. Line number 119 “Different nymphal instars of Aphids were used as a food source for both larvae and adults”.

Comment: Line number 83. Please provide thickness “Muslin cloth” 

Response: Done Accordingly. Line number 121 “muslin cloth (440 µm)”.

Comment: Line number 87. Provide authority name “Mustard aphid (Lipaphis erysimi), Citrus black aphid (Toxoptera Citricida), and Peach aphid (Diuraphis noxia)”

Response: Done Accordingly. Line number 124-125 “Mustard aphid (Lipaphis erysimi Kaltenbach), Citrus black aphid (Toxoptera citricida Kirkaldy), and Peach aphid (Diuraphis noxia Kurdjumov)

Comment: Line number 91. Please provide the temperature, RH and Photoperiod “under laboratory-controlled conditions”.

Response: Done Accordingly. Line number 129 “under laboratory-controlled conditions (26±2 °C, 65±5% R.H and L14:D10 photoperiod)”.

Comment: Line number 96. Please provide no of aphids released for each replication “The counted number of aphids were released in each replication”.

Response: Done accordingly. Line number 134 “The sufficient/similar number of aphids were released in each replication”.

Comment: Line number 97. Whether control treatment was maintained?

Response: This section has been revised and removed from the revised manuscript.

Comment: Line number 98-99. Please provide no of aphids released for each replication; Please indicate no of aphids provided for each adult “After the emergence of adults, five pairs (each pair represent one replication) of each treatment were placed in plastic jars”.

Response: As our objective was not to find out the predatory potential of C. sexmaculalata. Therefore we did not count number of aphids provided and provide sufficient/similar number of aphids in each replication.

Comment: Line number 127. please provide version of SAS 

Response: Done accordingly. Line number 171 “SAS (Version 8.0)”.

Comment: Line 165. “adult longevity, and female fecundity of C. sexmaculata”.

Response: Done accordingly. Line number 211 “adult longevity of C. sexmaculata”.

 

RESPONSE TO REVIEWER 2

Comment: The authors should add concluding lines. It should be more specific and striking. 

Response: Done accordingly.

Comment: The introduction section is not coherent. There are many useless sentences and lack key information. It should be rewritten completely. The author needs to add info about Age-stage, two-sex life table approach by TWO-SEX MS Chart. What is the difference between this technique and the traditional technique?

Response: Comprehensive rephrasing of introduction section has been done as suggested. Further, the traditional technique provide data related to female only while Age-stage two-sex life table provide data related to both male and female, that’s the reason of using this in the current research work. 

Comment: -In results, please follow the correct pattern for writing the statistical values. For example, in L168, you may replace (F=19.35; F2,87; P<0.0001) by (F=19.35, DF=2,87, P<0.0001). Please write the exact P value. Follow this pattern throughout the manuscript.

Response: Done accordingly throughout the manuscript.

Comment: It is recommended to discuss and explain what should be the appropriate policies based on the findings of the current study. Moreover, the results should be further elaborated to show how they could be used for real applications.

Response: Future recommendations is added in the conclusion section as recommended 

Comment: Figure 1-4: I strongly suggest to replace these figures by color figures. The colors must be more contrasting, so that readers can easily understand the variations among different parameters etc 

Response: Done accordingly. 

Comment: -In statistical analysis section, the authors missed key references (Chi et al. 2022a; 2022b) that strongly support the age-stage, two-sex, life table approach.

- Chi et al. 2022a. TWOSEX-MSChart: the key tool for life table research and education. Entomologia Generalis. 42 (6): 845-849.

- Chi et al. 2022b. Innovative application of set theory, Cartesian product, and multinomial theorem in demographic research. Entomologia Generalis. 42 (6) 863-874.

Response: Done accordingly.

Comment: - I strongly suggest the authors to check the population projection via TIMING-MSChart computer program. Construct figures of population projections, add formula of population projection statistical analysis section, add results with separate heading (such as Population Projection), and finally, discuss these results in discussion section by comparing it with recently published articles from reputable journals.

Response: As the current work was focused on life table parameters, population projection was not our objective that’s why data and as well as analysis was not aimed for population projection. But this useful suggestion is being incorporated in our future research work.

Comment: - I strongly suggest authors to check the predation rate using age-stage, two-sex life table approach, as this software precisely describes the predation rate.

Ding, H. Y., Lin, Y. Y., Tuan, S. J., Tang, L. C., Chi, H., Atlıhan, R., ... & Güncan, A. (2021). Integrating demography, predation rate, and computer simulation for evaluation of Orius strigicollis as biological control agent against Frankliniella intonsa. Entomologia Generalis, 41(2), 179-196.

- Islam, Y., Güncan, A., Fan, Y., Zhou, X., Naeem, A., & Shah, F. M. (2022). Age-stage, two-sex life table and predation parameters of Harmonia axyridis Pallas (Coleoptera: Coccinellidae), reared on Acyrthosiphon pisum (Harris)(Hemiptera: Aphididae), at four different temperatures. Crop Protection, 106029.

- Yılmaz, M., & Polat Akköprü, E. (2021). Predation rate linked to life table of Chrysoperla carnea (Stephen)(Neuroptera: Chrysopidae) fed on small walnut aphid (Chromaphis juglandicola)(Kalt.)(Hemiptera: Aphididae): with population and predation projections. Phytoparasitica, 49(2), 217-228.

Response: As the current work was focused on life table parameters and we did not record data related to the predatory potential. But we will work on your value able suggestion in our future projects. 

Comment: - In Table 2, the authors didn’t added the standard errors, also missed the P values and different letters to show the significant differences. These are very important, I am surprised why authors missed these key information.

Response: As suggested by the respected reviewer, table 2 is now revised by adding statistical values i.e., standard errors and letterings. 

Comment: 

Response: - The authors should add a separate conclusion section after discussion. The conclusion section should be concise and to the point.

Comment: Done accordingly.

Response:

Comment: Correct Ref# 39: “Chi H. TIMING-MSChart: a computer program for the population projection based on age-stage, two-sex life table. 2016” should be replace by “Chi, H. TWOSEX-MS Chart: A Computer Program for the Age-Stage, Two-Sex Life Table Analysis. 2022. Available online: http://140.120.197.173/ecology/Download/Twosex-MSChart-exe-B100000.rar (access date).” Please mention the access date (day, month, year). 

Response: Done accordingly.

Comment: Regarding data availability, upload all raw datasets of the life table and functional response as supplementary files. Raw data files should be available as supplementary files.

Response: The requested data has been as a supplementary file.

Comment: - Please improve the figures quality especially the font size and colors. These figures should be more striking.

Reponse: Done accordingly.

---

## [Editor Report · Decision Letter 1]

25 Jul 2023

Chemotaxis response and age-stage, two-sex life table of the Cheilomenes sexmaculata (Fabricius)  (coccinellidae: coleoptera) against different aphid species

PONE-D-22-16646R1

Dear Dr. Zaka,

We’re pleased to inform you that your manuscript has been judged scientifically suitable for publication and will be formally accepted for publication once it meets all outstanding technical requirements.

Kind regards,

Nicolas Desneux

Academic Editor

PLOS ONE
---

## [Editor Report · Acceptance letter]

26 Jan 2024

PONE-D-22-16646R1 

PLOS ONE

Dear Dr. Zaka, 

I'm pleased to inform you that your manuscript has been deemed suitable for publication in PLOS ONE. Congratulations! Your manuscript is now being handed over to our production team.

Kind regards, 

on behalf of

Dr. Nicolas Desneux 

Academic Editor

PLOS ONE